# The epidemiology of atopic dermatitis in older adults: A population-based study in the United Kingdom

Leslie N. Chan[1], Alexa Magyari[2], Morgan Ye[3], Noor A. Al-Alusi[1], Sinead M. Langan[4], David Margolis[5], Charles E. McCulloch[6], Katrina Abuabara[3]*

1 School of Medicine, University of California San Francisco, San Francisco, California, United States of America, 2 University of California Berkeley, Berkeley, California, United States of America, 3 Department of Dermatology, University of California San Francisco, San Francisco, California, United States of America, 4 Department of Non-communicable Disease Epidemiology, London School of Hygiene and Tropical Medicine, London, England, United Kingdom, 5 Department of Dermatology, Perelman School of Medicine, University of Pennsylvania, Philadelphia, Pennsylvania, United States of America, 6 Department of Epidemiology and Biostatistics, University of California San Francisco, San Francisco, California, United States of America

* katrina.abuabara@ucsf.edu

**Data Availability Statement:** The data underlying the results presented in the study are available from The Health Improvement Network. More information can be found at https://www.the-

## Abstract

### Background

Atopic dermatitis is known to be common among children, but there are few studies examining the epidemiology across the life course. In particular, there is a paucity of data on atopic dermatitis among older adults.

### Objective

To evaluate participant characteristics, patterns of disease activity and severity, and calendar trends in older adult atopic dermatitis in comparison to other age groups in a large population-based cohort.

### Methods

This was a cohort study of 9,154,936 individuals aged 0–99 years registered in The Health Improvement Network, a database comprised of electronic health records from general practices in the United Kingdom between 1994 and 2013. Atopic dermatitis was defined by a previously validated algorithm using a combination of at least one recorded atopic dermatitis diagnostic code in primary care and two atopic dermatitis therapies recorded on separate days. Cross-sectional analyses of disease prevalence were conducted at each age. Logistic mixed effect regression models were used to identify predictors of prevalent disease over time among children (0–17 years), adults (18–74 years), and older adults (75–99 years).

### Results

Physician-diagnosed atopic dermatitis was identified in 894,454 individuals with the following proportions in each age group: 18.3% of children, 7.7% of adults, and 11.6% of older adults. Additionally, atopic dermatitis prevalence increased across the 2-decade period

health-improvement-network.com/access-to-thin-data.

**Funding:** This research was specifically funded by grant number K23AR073915 from the National Institute of Arthritis and Musculoskeletal and Skin Diseases Career Development Award awarded to KA (https://www.niams.nih.gov/). SML receives funding by grant number 205039/Z/16/Z from a Wellcome Trust Senior Research Fellowship in Clinical Science (https://wellcome.org/). CEM received support from the National Center for Advancing Translational Sciences, National Institutes of Health, through UCSF-CTSI Grant Number KL2 TR001870 (https://ncats.nih.gov/). AM is supported by NIA grant T32-AG000246 (https://www.nia.nih.gov/). The findings and conclusions in this report are those of the authors and do not necessarily represent the views of the funders. The funders had no role in study design, data collection and analysis, decision to publish, or preparation of the manuscript.

**Competing interests:** I have read the journal's policy and the authors of this manuscript have the following competing interests: KA is a consultant for TARGET RWE, a company developing an atopic disease research registry, and her institution receives investigator-initiated grants from Pfizer and Cosmetique Active Internacional SNC. DJM receives funding to the University of Pennsylvania from Valeant for studies not directly related to this manuscript. He receives consulting funds from Leo, Sanofi, and Pfizer for topics related to atopic eczema but not this manuscript. No other disclosures were reported. This does not alter our adherence to PLOS ONE policies on sharing data and materials.

(beta from linear regression test for trend in the change in proportion per year = 0.005, p = 0.044). In older adults, atopic dermatitis was 27% less common among females (adjusted OR 0.73, 95% CI 0.70–0.76) and was more likely to be active (59.7%, 95% CI 59.5–59.9%) and of higher severity (mean annual percentage with moderate and severe disease: 31.8% and 3.0%, respectively) than in other age groups.

## Conclusion

In a large population-based cohort, the prevalence of physician-diagnosed atopic dermatitis has increased throughout adulthood and was most common among males age 75 years and above. Compared to children ages 0–17 and adults ages 18–74, older adult atopic dermatitis was more active and severe. Because the prevalence of atopic dermatitis among older adults has increased over time, additional characterization of disease triggers and mechanisms and targeted treatment recommendations are needed for this population.

## Introduction

Atopic dermatitis (also known as atopic eczema, or simply eczema) is a common inflammatory condition that manifests as dry, severely itchy skin [1]. While much of the early literature on this condition has focused on childhood disease, newer evidence suggests that atopic dermatitis is also common during adulthood [2, 3]. Older adults, in particular, have not been well-represented in epidemiologic studies of atopic dermatitis nor in clinical trials for novel treatments [4].

Differences in both demographic characteristics and clinical manifestations of atopic dermatitis have been identified between children and adults, demonstrating the importance of studying atopic dermatitis past childhood [2, 5, 6]. In a recent study to identify the prevalence of physician-diagnosed atopic dermatitis across the lifespan, we found unexpectedly high rates of atopic dermatitis among older adults [3]. A small body of literature has described 'senile' atopic dermatitis, but there remains a gap in our understanding of the epidemiology of atopic dermatitis in this population [6, 7]. Understanding disease trends among older adults is a priority because they comprise the most rapidly growing demographic group worldwide—by 2050, the population of adults over the age of 80 in the United States (US) is expected to triple [8]. Additionally, older adults are often affected by other health conditions that may influence atopic dermatitis treatment choice. These unique considerations underscore the importance of studying the epidemiology and disease characteristics of atopic dermatitis in the older adult population.

Our objective was to characterize the epidemiology of atopic dermatitis among older adults as compared to other age groups in a large population-based sample. Specifically, we examined the prevalence of atopic dermatitis by age and whether this changed over time, investigated the association between patient characteristics and the age-specific prevalence of atopic dermatitis, and identified patterns of disease activity and severity by age group.

## Methods

### Population

We conducted a longitudinal analysis using routinely collected primary care data from The Health Improvement Network (THIN) cohort (Fig 1). THIN consists of anonymized electronic health records of patients registered with more than 500 participating general practices

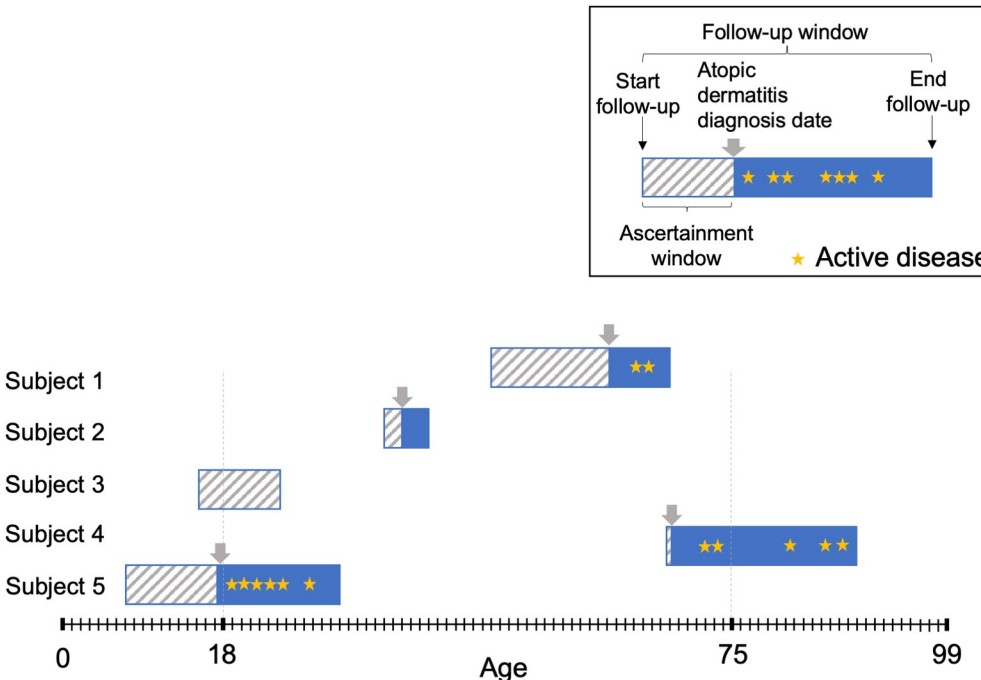

**Fig 1. Graphical depiction of study design.** Participants were followed for up to 20 years at differing ages; if individuals met atopic dermatitis diagnosis criteria, they were assessed for active disease during each subsequent year of follow-up.

in the United Kingdom (UK), covering approximately 6% of the UK population, and has been widely used for longitudinal analyses of chronic health conditions [9–13]. The data are valid for many chronic conditions, and the population is generalizable to the larger UK population in which general practitioners manage 97% of atopic dermatitis cases [10]. This study was approved by the THIN SRC (Reference 14–083), considered exempt by the University of Pennsylvania IRB, and not considered human subjects research by the UCSF IRB.

The analysis population included all patients between 0–99 years old who were registered at participating practices in the UK during the follow-up window. The start of follow-up began from the latest of the following dates: date the patient registered with the practice, date the practice met data quality control standards, or the start of the cohort (January 1, 1994). The follow-up period ended at the earliest of the following dates: recorded death date, last visit with the practice, end of registration with the practice, or the end of available follow-up data (January 15, 2013).

## Variables

Atopic dermatitis diagnosis was determined by a combination of at least one recorded atopic dermatitis diagnostic code in primary care (THIN, using Read codes) and two atopic dermatitis therapies recorded on separate days, as previously validated [9]. The positive predictive value of this algorithm for identifying physician-confirmed atopic dermatitis among all adults over age 18 was 82% (95% CI 73–89%). Among the subset of adults in this sample over age 75, the positive predictive value was 85% (95% CI 55–98%). Atopic dermatitis therapy categories included the following: emollients, topical corticosteroids, topical calcineurin inhibitors (tacrolimus and pimecrolimus), oral glucocorticoids, azathioprine, ciclosporin, methotrexate, and mycophenolate mofetil. The date of diagnosis was set as the date of the latest recorded

component (i.e. atopic dermatitis diagnostic code or second record for atopic dermatitis therapy) in the algorithm. Therapies were selected based on British National Formulary codes and individually reviewed to ensure relevance (i.e. topical rather than optical steroids) (S1 Table).

Prevalent atopic dermatitis was defined on a yearly basis as any additional atopic dermatitis medical code from a physician visit or prescription for an atopic dermatitis treatment during a year of follow-up after a patient met the atopic dermatitis definition. The year in which a participant ended their follow-up was used as a proxy to assess for the association between calendar time and atopic dermatitis. Socioeconomic status was measured using the Townsend Deprivation Score, an area-level measure encompassing unemployment and household crowding rates as well as home and car ownership, based on Census data for groups of about 150 households [14]. The score was reported in quintiles with higher quintiles reflecting higher amounts of deprivation, or lower socioeconomic status. Urban/rural classification, or setting, was based on the Office of National Statistics guidelines [15]. Demographic data were recorded at the time of registration.

## Analyses

Age-specific prevalence of atopic dermatitis was calculated in a series of cross-sectional analyses by dividing the number of patients who met the validated atopic dermatitis definition and had additional atopic dermatitis codes in any given year by the total number of patients being followed at the same age. Local polynomial smoothed plots were generated from cross-sectional calculations at each age by participant characteristics and by calendar period. Based on the demographic trends identified in these plots, we grouped the population into three age categories for stratified regression analyses: children (ages 0–17), adults (ages 18–74), and older adults (ages 75–99). Within each group, we quantified the association of age, calendar period, area of residence (urban/rural), sex, and Townsend Deprivation Score with the odds of prevalent atopic dermatitis during any given year using multivariable mixed effects logistic regression models. These variables were chosen *a priori* and were all included in the final models, regardless of significance level in univariate analyses.

We also examined atopic dermatitis activity and severity by age group. Disease activity was determined by dividing the number of years in which an individual had an additional diagnosis or treatment code for atopic dermatitis by the total number of years of follow-up for that individual. Disease severity was designated as mild, moderate, or severe based on treatment and referral patterns in a time-updated progressive manner. These designations were based on National Institute for Health and Care Excellence (NICE) treatment recommendations and have been used in prior studies [16, 17]. At any given point during follow-up, atopic dermatitis patients belonged to one of three severity categories: mild, moderate, or severe. By default, all individuals with atopic dermatitis were classified as having mild disease. They were classified as having moderate atopic dermatitis from the earliest of: 1) second potent topical steroid treatment within a year, or 2) first calcineurin inhibitor treatment (NICE recommends calcineurin inhibitors only as a second-line treatment for moderate-to-severe disease in the UK) [18]. Individuals were classified as having severe atopic dermatitis from the earliest of: 1) first systemic treatment for atopic dermatitis (i.e. a record of a prescription for cyclosporine, azathioprine, mycophenolate or methotrexate), or 2) first phototherapy code, or 3) first referral to secondary care for atopic dermatitis. Of note, only 3% of atopic dermatitis patients are managed by dermatologists in the UK, and referral to tertiary care is recommended for severe disease [19, 20]. Oral glucocorticoids were not included in the definition based on NICE guidance and international standards [18, 21]. Participants could progress from mild, to moderate, to severe disease but they did not necessarily have to progress in a stepwise order; they

could be assigned directly to the severe atopic dermatitis category. However, participants were not considered to progress to less severe categories of atopic dermatitis—once defined as moderate, atopic dermatitis patients remained as such unless they developed severe atopic dermatitis; once defined as severe, atopic dermatitis patients remained as such.

### Additional analyses

We performed additional analyses to examine the stability of our results to potential biases. First, to examine the possibility of bias due to short durations of follow-up, we repeated the calculation of disease activity after restricting to a subset of individuals with at least 2 years of follow-up. Second, we examined the possibility of ascertainment bias, given that young and old individuals are more likely to seek care and therefore may be more likely to receive atopic dermatitis diagnoses. In a random 1% sample of the population (N = 91,549), we recreated the graph depicting prevalence of disease by age after restricting the sample to individuals with at least one physician's visit at that age.

Finally, we recreated the same graph in a validation analysis using data from the 2005–2006 US National Health and Nutrition Examination Survey (NHANES). The NHANES 2005–2006 cohort was selected from among other waves of NHANES because it contained several questions specifically pertaining to atopic dermatitis and was administered to individuals of all ages (0–85+), demonstrating the greatest overlap in age with our sample in THIN. We combined questions as others have done [22] and examined three potential definitions related to atopic dermatitis: 1) Answered 'yes' to "Ever had an itchy rash which was coming and going for at least 6 months?" and "Itchy rash at any time in the last 12 months?" 2) Answered 'yes' to Definition 1 questions, and answered 'yes' to "Has this rash cleared up completely at any time during the last 12 months?" and "Itchy rash at any time affecting: the folds of the elbows, behind the knees, in front of the ankles, under the buttocks, or around the neck, ears, or eyes?" 3) Answered 'yes' to Definition 1 questions, and answered 'yes' to "Doctor or health professional ever told you that you had eczema?" An explanation for the measure of socioeconomic status used in this validation analysis is included in S1 Methods.

### Missing data

We examined patterns of missing data by age and atopic dermatitis status (S2 and S3 Tables). For the primary analysis, we performed a complete case analysis. To explore the effect of missing data, we performed multiple imputation with iterative chained equations to impute missing covariate data in setting and Townsend Score. Given the large number of observations in this dataset and the nature of repeated measures, we performed multiple imputation on a random 1% sample of the population (N = 91,549). Ten imputed datasets were generated, and the average results from repeated analyses were compared with complete case analysis in the same random 1% sample. All statistical analyses were performed using Stata 16.1 (StataCorp).

## Results

### Participant characteristics

We identified 894,454 individuals who ever met the definition for atopic dermatitis among a total population of 9,154,936. On average, a greater proportion of participants with atopic dermatitis was female, resided in an urban setting, and had higher socioeconomic status compared with non-atopic dermatitis patients (Table 1). Overall, 8.3% of participants were missing covariate data on the Townsend Deprivation Score and 20.9% were missing data on urban/rural setting. This proportion with missing data were similar between age group, though

**Table 1. Participant characteristics.**

| Characteristic | All (N = 9,154,936) | Atopic dermatitis (N = 894,454) | Non-atopic dermatitis (N = 8,260,482) |
|---|---|---|---|
| Age at start of follow-up, No. (%) | | | |
| Children (0–17 years) | 1,400,354 | 255,622 (18.3) | 1,144,732 (81.7) |
| Adults (18–74 years) | 6,731,678 | 520,402 (7.7) | 6,211,276 (92.3) |
| Older adults (75–99 years) | 1,022,904 | 118,430 (11.6) | 904,474 (88.4) |
| Sex, No. (%) | | | |
| Male | 4,443,260 | 396,760 (8.9) | 4,046,500 (91.1) |
| Female | 4,711,676 | 497,694 (10.6) | 4,213,982 (89.4) |
| Setting, No. (%) | | | |
| Urban | 5,906,148 | 602,551 (10.2) | 5,303,597 (89.8) |
| Rural | 1,334,062 | 138,072 (10.3) | 1,195,990 (89.7) |
| Missing | 1,914,726 | 153,831 (8.0) | 1,760,895 (92.0) |
| Townsend score, No. (%) | | | |
| 1 (least deprived) | 1,887,796 | 215,341 (11.4) | 1,672,455 (88.6) |
| 2 | 1,687,078 | 175,969 (10.4) | 1,511,109 (89.6) |
| 3 | 1,771,062 | 172,697 (9.8) | 1,598,365 (90.2) |
| 4 | 1,745,740 | 159,622 (9.1) | 1,586,118 (90.9) |
| 5 (most deprived) | 1,307,708 | 116,183 (8.9) | 1,191,525(91.1) |
| Missing | 755,552 | 54,642 (7.2) | 700,910 (92.8) |
| Follow-up time | | | |
| Mean years (SD) | 7.6 (6.4) | 10.7 (6.3) | 7.3 (6.3) |

participants with missing data were less likely to have atopic dermatitis diagnoses (Table 1, S2 and S3 Tables).

## Atopic dermatitis prevalence by participant characteristic

Annual atopic dermatitis prevalence ranged by age from 6.1–21.2% among children ages 0–17, 3.5–7.8% among adults ages 18–74, and 7.0–9.3% among older adults ages 75–99. In mixed models, we found that the prevalence of atopic dermatitis increased with age among older adults (Adjusted Odds Ratio (AOR) 1.06, 95% CI 1.06–1.06, i.e. a 6% increase in odds per year of age), whereas the prevalence decreased by 14% annually among children (Fig 2 and Table 2).

Atopic dermatitis was less common among female older adults: they had about three-fourths the odds of prevalent disease as their male counterparts (AOR 0.73, 95% CI 0.70–0.76). In contrast, female adults ages 18–74 years had a nearly three times higher odds of atopic dermatitis (AOR 2.72, 95% CI 2.68–2.76) compared to males in their age group. The odds of atopic dermatitis were 9–24% higher in urban as compared to rural settings across the age categories. Socioeconomic trends differed only among children: the odds of atopic dermatitis were smaller in more deprived groups among those 0–17 years, though in adults ages 18–74 years and 75–99 years there was no difference between the most deprived and least deprived groups, though the intermediately deprived groups (Townsend 2–4) still had higher odds of atopic dermatitis compared to the least deprived group (Table 2). Analyses in a random 1% sample of the population showed similar results when performing complete case analysis compared to multiple imputation (S4 Table).

A sensitivity analysis restricted to individuals with at least one physician's visit during that year did not show evidence of ascertainment bias; although the prevalence was slightly higher at all ages, the shape of the curve was similar (S1 Fig).

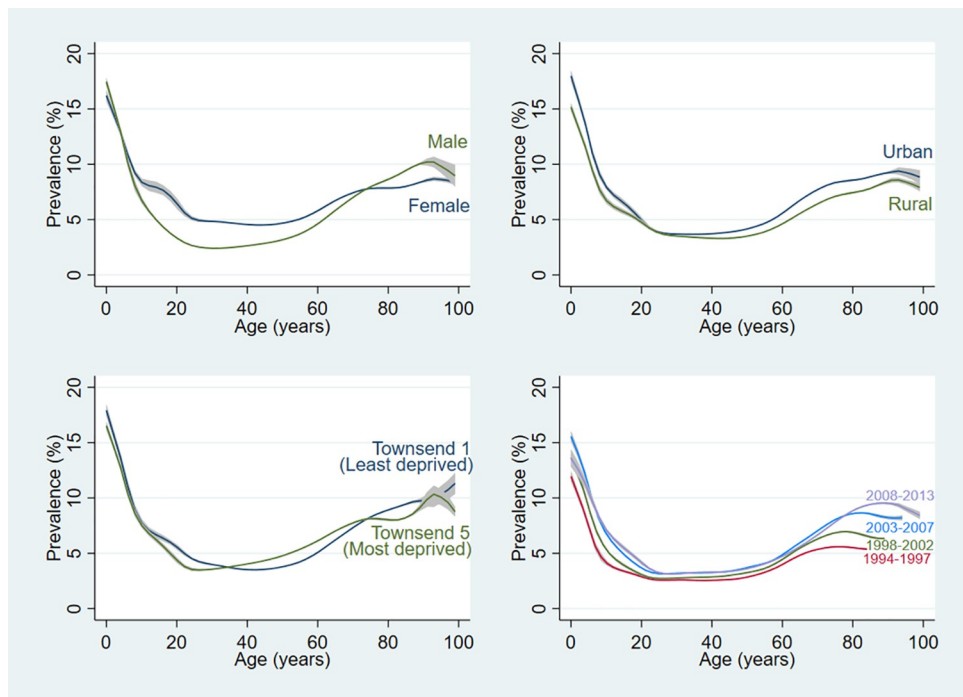

**Fig 2. Annual period prevalence of atopic dermatitis by participant characteristic and calendar period.** Local polynomial smoothed plots with shading indicating the 95% CIs generated from yearly cross-sectional calculations of the proportion of participants with prevalent atopic dermatitis from ages 0 to 99.

In a validation analysis using NHANES, a US population-based survey, we found a qualitatively similar trend in the sense that atopic dermatitis prevalence increased across adulthood—4.5–20.7% reported symptoms in the past year among those ages 75 and older

**Table 2. Multivariable mixed effects logistic regression analysis of atopic dermatitis by age group, according to participant characteristic.**

| Variable | Age Group | | |
|---|---|---|---|
| | **0–17 Years (N = 1,959,250)** | **18–74 Years (N = 5,603,315)** | **75–99 Years (N = 650,400)** |
| | AOR (95% CI) from regression model | | |
| Age[a] | 0.86 (0.86, 0.86) | 1.04 (1.03, 1.04) | 1.06 (1.06, 1.06) |
| Calendar year at end of follow-up[a] | 1.12 (1.11, 1.12) | 3.46 (3.40, 3.52) | 1.07 (1.07, 1.08) |
| Sex | | | |
| Male | Reference | | |
| Female | 1.16 (1.14, 1.18) | 2.72 (2.68, 2.76) | 0.73 (0.70, 0.76) |
| Setting | | | |
| Rural | Reference | | |
| Urban | 1.24 (1.21, 1.26) | 1.13 (1.11, 1.15) | 1.09 (1.06, 1.14) |
| Townsend score | | | |
| Townsend 1 (least deprived) | Reference | | |
| Townsend 2 | 0.94 (0.92, 0.96) | 0.95 (0.94, 0.97) | 0.92 (0.87, 0.97) |
| Townsend 3 | 0.90 (0.88, 0.92) | 0.95 (0.93, 0.97) | 0.84 (0.80, 0.89) |
| Townsend 4 | 0.85 (0.83, 0.87) | 0.95 (0.93, 0.97) | 0.87 (0.82, 0.92) |
| Townsend 5 (most deprived) | 0.90 (0.88, 0.93) | 1.02 (1.00, 1.05) | 0.97 (0.91, 1.04) |

*Note.* Each column represents a separate multivariable mixed effects logistic regression model. AOR = adjusted odds ratio.

[a] Modeled as a continuous variable, value represents adjusted odds of atopic dermatitis for each additional year of age or each additional calendar year, respectively.

(S2 Fig). Similar to our primary analysis, atopic dermatitis was also more common among male older adults.

## Atopic dermatitis activity

Atopic dermatitis activity was highest in older adults (59.7%, 95% CI 59.5–59.9%), followed by children (48.4%, 95% CI 48.3–48.5%) then adults (42.2%, 95% CI 42.1–42.3%) (S3 Fig). The results were similar after restricting to participants with at least 2 years of follow-up time in a sensitivity analysis.

## Atopic dermatitis severity

Across all ages, the majority of atopic dermatitis patients had mild disease (mean annual prevalence in children: 92.5%, adults: 77.1%, and older adults: 65.2%). The mean annual percentage of atopic dermatitis patients with moderate and severe disease was highest among older adults 31.8% and 3.0%, respectively (Fig 3). By the end of follow-up, 46.2% of older adults met the definition of moderate disease and 5.2% met the definition of severe disease. As described in the methods, oral glucocorticoids were not included in the severity definition. The proportion of individuals who received an oral glucocorticoid prescription (for any reason) was 8.7% of those designated as having mild atopic dermatitis at the end of follow-up, 15.4% of those with moderate atopic dermatitis, and 18.8% of those with severe atopic dermatitis.

## Calendar trends

The total proportion of individuals with atopic dermatitis increased with calendar time (beta from linear regression test for trend in the change in proportion per year = 0.005, p = 0.044, S5 Table). In the most recent time span of 2008 to 2013, the increase in percent with atopic

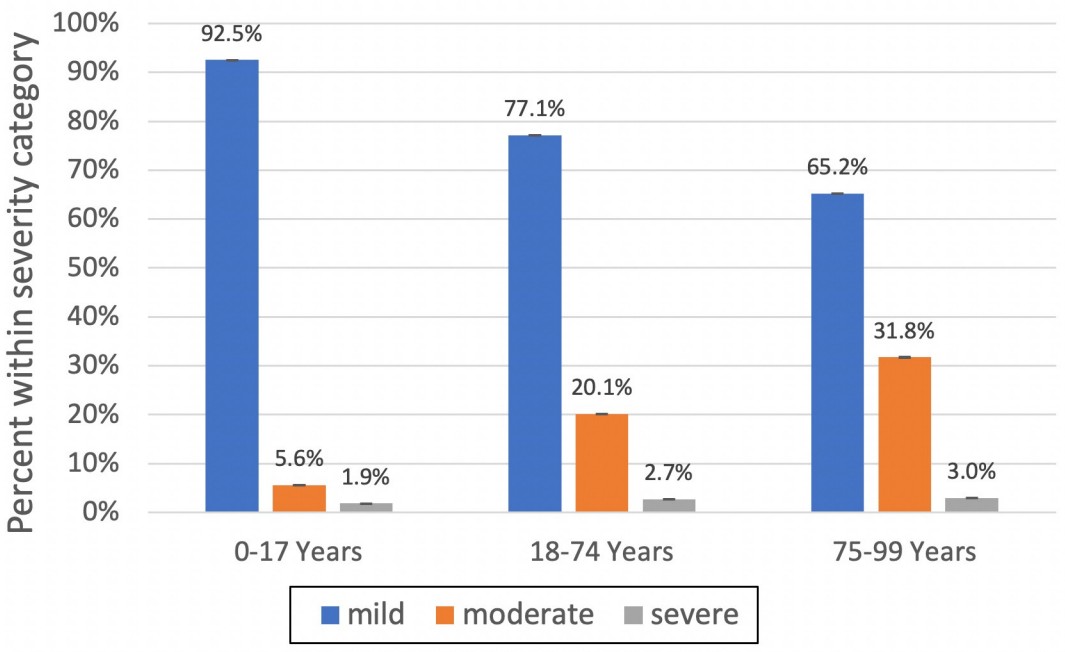

**Fig 3. Atopic dermatitis severity, by age group.** Mean percentage of patients in each age group (children, adults, and older adults) that meet the definition for mild, moderate, and severe atopic dermatitis. Bars indicate 95% confidence intervals.

dermatitis plateaued in children and adults but continued to rise in older adults (Fig 2). Additionally, in the mixed models, odds of atopic dermatitis increased with calendar time, and the annual effect was strongest in adults ages 18–74 years (AOR 3.46, 95% CI 3.40–3.52) (Table 2).

## Discussion

In a population-based cohort of over 9 million individuals from the UK, we found that the prevalence of adult atopic dermatitis increased over time and with age. In comparison to children ages 0–17 and adults ages 18–74, atopic dermatitis among older adults ages 75–99 years was active for a greater proportion of the follow-up time, more likely to be moderate or severe, and more likely to occur in men.

Although there are a few other studies that focus specifically on the epidemiology of atopic dermatitis among older adults, review of additional population-based cohorts, including the US NHANES validation cohort presented herein, the 2010 US National Health Interview Survey (NHIS) data [23], and two additional UK-based sources [24, 25] support our findings. The period prevalence of self-reported atopic dermatitis in the 2010 NHIS was similar to our findings: 9.0% among younger adults ages 18–32 and 11.4% among older adults ages 62–85. Our findings are also consistent with published literature on calendar trends in children and younger adults: a systematic review and meta-analysis found there was a >10% increase in lifetime prevalence of atopic dermatitis in the UK between 1990 and 2010 [26]. However, only one of 69 studies from this review included older adults from the time span of 2001–2005; our study builds on this literature by including data on a large population of older adults over a 20-year period.

Patient characteristics differed between age groups. For instance, similar to others, we found that atopic dermatitis was more than twice as common among females compared to males in the 18–74 age group [27–29]. However, our data suggest it is more common among males compared to females in the 75–99 age group, which is consistent with data from studies of smaller clinical population [30–32]. The reason for this is unclear; other authors have hypothesized that higher average estradiol levels in older men compared to older women may impact the production of cytokines that lead to the predominance of atopic dermatitis seen in older adult males [7]. In terms of socioeconomic status (SES), similar to older reviews, our study found a positive association of atopic dermatitis with less deprived or higher SES in children, whereas in adults the association was less pronounced. A systematic review and meta-analysis found that atopic dermatitis is more common among groups with higher SES, though these were primarily studies of children [33]. More recent studies have found that this association was strongest early in life, but that atopic dermatitis was more common among lower socioeconomic groups in adulthood or did not vary by SES [5, 25, 34]. Additional study is warranted to explore possible mechanisms for these findings. Finally, atopic dermatitis was more common in urban areas at all ages, a finding that may relate to lifestyle or environmental characteristics such as the types of microbes or increased pollution encountered in urban settings [35, 36].

Limitations of our study include the possibility for misclassification bias, which may be higher in adults than children, given a more heterogeneous presentation and a greater chance for other common adult skin conditions to overlap with or be misdiagnosed as atopic dermatitis. We used a validated algorithm of physician diagnoses with good positive predictive value for both children and adults in this cohort [9], and a previous study using this algorithm showed similar prevalence curves even after adjusting for potential misdiagnosis of conditions on the differential diagnosis of atopic dermatitis (e.g. psoriasis, contact dermatitis, scabies, seborrheic dermatitis, and drug- or photo-induced dermatitis) [3]. Another limitation relates to

the lack of detailed measures of disease activity, severity, and clinical presentation. We relied on time-updated electronic medical record-based measures of disease activity and severity because a large population-based cohort is needed to establish epidemiologic trends. While more detailed data would be useful to additionally characterize atopic dermatitis among older adults, it is likely to come from smaller and less representative populations. To address issues with missing data, we performed multiple imputation in a random 1% sample of the population and found the results consistent with those of complete case analysis in this sample. Finally, although we had standardized data on participants ages 0–99, the mean duration of follow-up was only 7.6 years, and as such we are unable to track an individual's disease severity and activity across the lifespan. Additional work is needed to further elucidate individual changes in disease activity and severity across age groups.

Of note, in our validation analysis using NHANES data, the trend of increasing prevalence among older adults was sensitive to the specific question used–the prevalence did not increase among those also reporting flexural rashes and physician diagnosis. This could be because these questions were designed to detect atopic dermatitis in children and younger adults, who more commonly present with flexural disease and may be more likely to be recognized and diagnosed by physicians [6, 7]. Future research should examine the validity of diagnostic criteria among older adults and regional variations in physician diagnosis and terminology.

There are a number of reasons why additional study of atopic dermatitis among older adults is important. First, older adults are the most rapidly increasing demographic segment [8], and atopic dermatitis appears to be increasing among this group. Second, older adults are at greater risk for a number of comorbid conditions that may be more common among individuals with atopic dermatitis, including depression and anxiety, cardiovascular disease, osteoporosis, and dementia [37–39]. Third, the clinical presentation and underlying pathogenesis may differ among this age group. Previous studies have found that atopic dermatitis presents in a different distribution on the body in older adults—instead of the classic flexural lichenification found in children and younger adults, atopic dermatitis in older adults more often has reverse signs of lichenification around unaffected elbow and knee folds and higher incidence on the buttocks and genitals instead of on the face and scalp [7, 40]. Moreover, older adults may have other common pruritic skin conditions that must be ruled out prior to the diagnosis of atopic dermatitis. Additionally, they may also have other coexisting health conditions and are more likely to take systemic medications that may cause pruritus and rash that could be mistaken for atopic dermatitis [7]. Additional research is needed to refine diagnostic criteria, assess detailed clinical characteristics, ascertain the inflammatory profile and biomarkers in this population, and identify disease triggers. Finally, this population is less likely to be represented in clinical trials [41], and should be considered for inclusion in testing new small molecules and biologic agents for atopic dermatitis.

## Supporting information

**S1 Fig. Correcting for ascertainment bias.** Local polynomial smoothed plots with shading indicating the 95% CIs generated from yearly cross-sectional calculations of the prevalence of atopic dermatitis from ages 0 to 99 within a 1% random sample of the population. In a correction for ascertainment bias, the prevalence of atopic dermatitis by year of age was restricted to those who had at least one medical code corresponding to a physician's visit during that year. (PDF)

**S2 Fig. Prevalence of atopic dermatitis across age in the 2005–2006 NHANES.** Local polynomial smoothed plots with shading indicating the 95% CIs generated from yearly cross-sectional calculations of the percent with prevalent atopic dermatitis during the past year from

ages 0 to 85+. A) Three definitions representing prevalence of atopic dermatitis were formed by separate combinations of survey questions relevant to atopic dermatitis B) Prevalence of atopic dermatitis by sex using Definition 1 and C) Prevalence of atopic dermatitis by income using Poverty Income Ratio (PIR) using Definition 1.
(PDF)

**S3 Fig. Atopic dermatitis activity, by age group.** Mean proportion of years with prevalent atopic dermatitis as a percent of total years of follow-up 95% confidence interval by age group. Primary analysis includes atopic dermatitis activity regardless of the number of years of follow-up. Sensitivity analysis excludes those with less than 2 years of follow-up time (total of 2.3% of individuals across all age groups).
(PDF)

**S1 Table. Atopic dermatitis diagnosis and therapy codes.**
(PDF)

**S2 Table. Proportion of missing data for setting by age group.**
(PDF)

**S3 Table. Proportion of missing data for Townsend score by age group.**
(PDF)

**S4 Table. Mixed effects logistic regression results for complete case analysis and multiple imputation in random 1% sample of population.**
(PDF)

**S5 Table. Mean percent of atopic dermatitis across age groups by 5-year time spans from 1994–2013.**
(PDF)

**S1 Methods. Socioeconomic status in the NHANES validation analysis.**
(PDF)

## Author Contributions

**Conceptualization:** Leslie N. Chan, Katrina Abuabara.

**Data curation:** Alexa Magyari, Morgan Ye.

**Formal analysis:** Leslie N. Chan, Alexa Magyari.

**Funding acquisition:** Katrina Abuabara.

**Investigation:** Leslie N. Chan.

**Methodology:** Leslie N. Chan, Alexa Magyari, Morgan Ye, Noor A. Al-Alusi, Charles E. McCulloch, Katrina Abuabara.

**Project administration:** Katrina Abuabara.

**Resources:** Alexa Magyari, Morgan Ye, Katrina Abuabara.

**Software:** Alexa Magyari, Morgan Ye.

**Supervision:** Katrina Abuabara.

**Validation:** Leslie N. Chan.

**Visualization:** Leslie N. Chan.

**Writing – original draft:** Leslie N. Chan, Katrina Abuabara.

**Writing – review & editing:** Leslie N. Chan, Alexa Magyari, Morgan Ye, Noor A. Al-Alusi, Sinead M. Langan, David Margolis, Charles E. McCulloch, Katrina Abuabara.

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
