## [Decision Letter · Decision Letter 0]

18 Jun 2021

PONE-D-21-12334

The epidemiology of atopic eczema in older adults: A population-based study in the United Kingdom

PLOS ONE

Dear Dr. Abuabara,

Thank you for submitting your manuscript to PLOS ONE. After careful consideration, we feel that it has merit but does not fully meet PLOS ONE’s publication criteria as it currently stands. Therefore, we invite you to submit a revised version of the manuscript that addresses the points raised during the review process.

We look forward to receiving your revised manuscript.

Kind regards,

Dong Keon Yon, MD, FACAAI

Academic Editor

PLOS ONE

Journal Requirements:

I have read the journal's policy and the authors of this manuscript have the following competing interests: KA is a consultant for TARGET RWE, a company developing an atopic disease research registry, and her institution receives investigator-initiated grants from Pfizer. DJM receives funding to the University of Pennsylvania from Valeant for studies not directly related to this manuscript. He receives consulting funds from Leo, Sanofi, and Pfizer for topics related to atopic eczema but not this manuscript. No other disclosures were reported.

Additional Editor Comments:

Thank you for submitting your manuscript “The epidemiology of atopic eczema in older adults: A population-based study in the United Kingdom” to Plos One. The reviewers and I believe it is of potential value for our readers. However, the reviewers have raised a number of very important issues, esp. R1-2, and their excellent comments will need to be adequately addressed in a revision before the acceptability of your manuscript for publication in the Journal can be determined.

Also I have some minor comments.

#1. Although your group have published the top tier journal using the terms atopic eczema, atopic dermatitis is the most commonly used term.

Kantor R, Thyssen JP, Paller AS, Silverberg JI. Atopic dermatitis, atopic eczema, or eczema? A systematic review, meta-analysis, and recommendation for uniform use of 'atopic dermatitis'. Allergy. 2016 Oct;71(10):1480-5. doi: 10.1111/all.12982. Epub 2016 Aug 3. PMID: 27392131; PMCID: PMC5228598.

#2. In abstract, please describe the AD algorithm briefly to reduce the reading burden for our readers.

#3. In abstract results, describe more information like Global Burden Disease paper or Lancet style.

Reviewers' comments:

Reviewer's Responses to Questions

**Comments to the Author**

1. Is the manuscript technically sound, and do the data support the conclusions?

Reviewer #1: Yes

Reviewer #2: Yes

2. Has the statistical analysis been performed appropriately and rigorously? 

Reviewer #1: Yes

Reviewer #2: Yes

3. Have the authors made all data underlying the findings in their manuscript fully available?

Reviewer #1: Yes

Reviewer #2: Yes

4. Is the manuscript presented in an intelligible fashion and written in standard English?

Reviewer #1: Yes

Reviewer #2: Yes

5. Review Comments to the Author

Reviewer #1: Dear authors

The subject is interesting and the text is well written, but the conclusions are poor and do not fully respond to the proposed objectives. Some references need to be adequate within the journal's norms.

I think that manuscript is adequate for publication after the adjustments.

Reviewer #2: This is an interesting and well-designed study that addresses the prevalence and characteristics of atopic eczema in the elderly, using a large population based database in the UK. The study reports that physician-diagnosed atopic eczema is highly prevalent in the older adult population, and has higher activity and severity than in other age groups. The manuscript is methodologically sound and well-written. It has no major flaws, but there are some minor issues to consider:

1. (Page 5 line 113) It would be helpful to the reader and other researchers to provide the full code list of atopic eczema diagnoses and therapies as a supplement.

2. (Page 7 line 152) While it is agreeable that the use of oral corticosteroids is not included in the definition of mild/moderate/severe eczema, it is an important confounding factor in atopic eczema symptoms. Please address this issue and if possible provide percentage of patients using oral corticosteroids within each severity group.

3. (Page 9 line 187, 195, Page11 Table 2) A substantial proportion of participants (8.3% on urban/rural setting, and 20.9% on Townsend Deprivation Score) have missing data. If the data is considered missing at random, has multiple imputation been considered? Also, the participants with missing Townsend Scores are less likely to have atopic eczema. This may influence the association between Townsend Scores and eczema prevalence for each age group. Please provide data on missing values and eczema prevalence for each age group.

4. (Page15 line 297) The algorithm to define atopic eczema used in this study has been validated in a prior study (Abuabara K, et al., Invest Dermatol. 358 2017;137(8):1655–62.) in children and adults, but has this been validated in the elderly (75-99) as well?

6. PLOS authors have the option to publish the peer review history of their article (what does this mean?). If published, this will include your full peer review and any attached files.

Reviewer #1: **Yes: **Marilda Aparecida Milanez Morgado de Abreu

Reviewer #2: No

---

## [Author Response · Author response to Decision Letter 0]

16 Aug 2021

Katrina Abuabara, MD, MA, MSCE

Associate Professor of Dermatology

 Associate Adjunct Professor of 

Epidemiology (UC Berkeley) 

2340 Sutter Street, N421

San Francisco, CA 94143-0808

katrina.abuabara@ucsf.edu

 

Editor

We ensured that our manuscript meets PLOS ONE’s style requirements and file naming guidelines. 

I have read the journal's policy and the authors of this manuscript have the following competing interests: KA is a consultant for TARGET RWE, a company developing an atopic disease research registry, and her institution receives investigator-initiated grants from Pfizer. DJM receives funding to the University of Pennsylvania from Valeant for studies not directly related to this manuscript. He receives consulting funds from Leo, Sanofi, and Pfizer for topics related to atopic eczema but not this manuscript. No other disclosures were reported.

Please confirm that this does not alter your adherence to all PLOS ONE policies on sharing data and materials, by including the following statement: "This does not alter our adherence to PLOS ONE policies on sharing data and materials.”

We have added this revised statement to our cover letter and confirmed with our co-authors that there are no restrictions on sharing of data and/or materials. 

Thank you for pointing this out. We have removed the phrase “data not shown” and instead added detailed tables on missing data in the Supporting Information (S1 Table and S2 Table).

We have ensured that the references are accurate, none have been retracted, and they are in the proper citation style. We noted articles by de Lusignan et al. and Blak et al. were listed twice, which we corrected. Online reports were also updated to the correct citation format. We also replaced one citation (Bieber T. Atopic Dermatitis. N Engl J Med. 2008 Apr 3;358(14):1483–94) with a more updated review article (Langan SM, Irvine AD, Weidinger S. Atopic dermatitis. Lancet. 2020 Aug 1;396(10247):345–60). We added one citation (Asher MI, Keil U, Anderson HR, Beasley R, Crane J, Martinez F, et al. International Study of Asthma and Allergies in Childhood (ISAAC): rationale and methods. Eur Respir J. 1995;8(3):483–91.) to provide more clarity for our additional analysis. 

5. Although your group have published the top tier journal using the terms atopic eczema, atopic dermatitis is the most commonly used term.

Kantor R, Thyssen JP, Paller AS, Silverberg JI. Atopic dermatitis, atopic eczema, or eczema? A systematic review, meta-analysis, and recommendation for uniform use of 'atopic dermatitis'. Allergy. 2016 Oct;71(10):1480-5. doi: 10.1111/all.12982. Epub 2016 Aug 3. PMID: 27392131; PMCID: PMC5228598.

Unfortunately, there is a lack of consensus on terminology in this field. A more recent citation by some of the same authors using survey data from the International Eczema Council concluded that “atopic eczema” is also a recommended term (Silverberg JI, Thyssen JP, Paller AS, Drucker AM, Wollenberg A, Lee KH, Kabashima K, Todd G, Schmid-Grendelmeier P, Bieber T. What's in a name? Atopic dermatitis or atopic eczema, but not eczema alone. Allergy. 2017 Dec;72(12):2026-2030. doi: 10.1111/all.13225. Epub 2017 Jun 30. PMID: 28605026). We had chosen atopic eczema to reflect the more common terminology used in the UK, where the cohort is based. Nonetheless, to facilitate consistency within the journal, we have changed “atopic eczema” to “atopic dermatitis” throughout the manuscript and figures/tables. 

6. In abstract, please describe the AD algorithm briefly to reduce the reading burden for our readers.

We have included a description of the AD algorithm in the Methods subsection of the abstract as follows: “Atopic dermatitis was defined by a previously validated algorithm using a combination of at least one recorded atopic dermatitis diagnostic code in primary care and two atopic dermatitis therapies recorded on separate days.”

7. In abstract results, describe more information like Global Burden Disease paper or Lancet style.

Thank you for your suggestion. We have described more information in our abstract results section as follows: 

“Results: Physician-diagnosed atopic dermatitis was identified in 894,454 individuals with the following proportions in each age group: 18.3% of children, 7.7% of adults, and 11.6% of older adults. Additionally, atopic dermatitis prevalence increased across the 2-decade period (beta from linear regression test for trend in the change in proportion per year = 0.005, p=0.044). In older adults, atopic dermatitis was 27% less common among females (adjusted OR 0.73, 95% CI 0.70-0.76) and was more likely to be active (59.7%, 95% CI 59.5-59.9%) and of higher severity (mean annual percentage with moderate and severe disease: 31.8% and 3.0%, respectively) than in other age groups.”

Reviewer #1

1. The subject is interesting and the text is well written, but the conclusions are poor and do not fully respond to the proposed objectives.

We reviewed the conclusions to ensure they align directly with the objectives and added more detail to fully convey the significance and future directions to readers as follows:

“In a large population-based cohort, the prevalence of physician-diagnosed atopic dermatitis in adulthood has increased over time and was found to be most common in males among those 75 years and older. Compared to children ages 0-17 and adults ages 18-74, atopic dermatitis in older adults ages 75-99 was also more active and severe. Because the prevalence of atopic dermatitis among older adults has increased over time, additional characterization of disease triggers and mechanisms and targeted treatment recommendations are needed for this population.”

2. Some references need to be adequate within the journal's norms.

Thank you for pointing this out. As described above, we have reviewed the references and corrected those that were not within the journal’s norms.

Reviewer #2:

1. (Page 5 line 113) It would be helpful to the reader and other researchers to provide the full code list of atopic eczema diagnoses and therapies as a supplement.

Thank you for your suggestion. We have included the full code list for atopic dermatitis diagnoses and therapies in the Supporting Information (S1 Table).

2. (Page 7 line 152) While it is agreeable that the use of oral corticosteroids is not included in the definition of mild/moderate/severe eczema, it is an important confounding factor in atopic eczema symptoms. Please address this issue and if possible provide percentage of patients using oral corticosteroids within each severity group.

Thank you for your comment. We added the proportion of individuals who received systemic corticosteroids (oral or injectable) in each severity group to the Results section of the manuscript as follows: 

“The proportion of individuals who received an oral glucocorticoid prescription (for any reason) was 8.7% of those designated as having mild atopic dermatitis at the end of follow-up, 15.4% of those with moderate atopic dermatitis, and 18.8% of those with severe atopic dermatitis.”

If a patient with atopic dermatitis were to receive only systemic corticosteroids as treatment in any given year, it is possible that we may underestimate disease activity. This is unlikely to serve as an important confounder, however, given that our definition of disease activity also included medical codes and other treatment codes. 

3. (Page 9 line 187, 195, Page11 Table 2) A substantial proportion of participants (8.3% on urban/rural setting, and 20.9% on Townsend Deprivation Score) have missing data. If the data is considered missing at random, has multiple imputation been considered? Also, the participants with missing Townsend Scores are less likely to have atopic eczema. This may influence the association between Townsend Scores and eczema prevalence for each age group. Please provide data on missing values and eczema prevalence for each age group.

We have provided data on missing values and eczema prevalence for each age group in the Supporting Information (S2 Table and S3 Table). To address issues surrounding missing data, we performed multiple imputation using a random 1% sample of the population. The results were consistent between the multiple imputation and complete case analysis and have included these results in the Supporting Information (S4 Table). 

4. (Page15 line 297) The algorithm to define atopic eczema used in this study has been validated in a prior study (Abuabara K, et al., Invest Dermatol. 358 2017;137(8):1655–62.) in children and adults, but has this been validated in the elderly (75-99) as well?

Thank you for the opportunity to clarify. Yes, this algorithm has been validated in ages 75-99 as well. We added the PPV for this age group to the Variables subsection of the Methods for clarity as follows: 

“The positive predictive value of this algorithm for identifying physician-confirmed atopic dermatitis among all adults over age 18 was 82% (95% CI 73-89%). Among the subset of adults in this sample over age 75, the positive predictive value was 85% (95% CI 55-98%).”

---

## [Decision Letter · Decision Letter 1]

22 Sep 2021

The epidemiology of atopic dermatitis in older adults: A population-based study in the United Kingdom

PONE-D-21-12334R1

Dear Dr. Abuabara,

We’re pleased to inform you that your manuscript has been judged scientifically suitable for publication and will be formally accepted for publication once it meets all outstanding technical requirements.

Kind regards,

Dong Keon Yon, MD, FACAAI

Academic Editor

PLOS ONE

Additional Editor Comments (optional):

I congratulate you on this mesmerizing work.

Reviewers' comments:

Reviewer's Responses to Questions

**Comments to the Author**

1. If the authors have adequately addressed your comments raised in a previous round of review and you feel that this manuscript is now acceptable for publication, you may indicate that here to bypass the “Comments to the Author” section, enter your conflict of interest statement in the “Confidential to Editor” section, and submit your "Accept" recommendation.

Reviewer #1: All comments have been addressed

Reviewer #2: All comments have been addressed

2. Is the manuscript technically sound, and do the data support the conclusions?

Reviewer #1: Yes

Reviewer #2: Yes

3. Has the statistical analysis been performed appropriately and rigorously? 

Reviewer #1: Yes

Reviewer #2: Yes

4. Have the authors made all data underlying the findings in their manuscript fully available?

Reviewer #1: Yes

Reviewer #2: Yes

5. Is the manuscript presented in an intelligible fashion and written in standard English?

Reviewer #1: Yes

Reviewer #2: Yes

6. Review Comments to the Author

Reviewer #1: The authors have adequately addressed my comments raised in a previous round of review.

This manuscript is now acceptable for publication.

Reviewer #2: The authors have thoroughly responded to all the comments and made corrections in the revised manuscript. In my view, the manuscript is acceptable for publication in the current state.

7. PLOS authors have the option to publish the peer review history of their article (what does this mean?). If published, this will include your full peer review and any attached files.

Reviewer #1: No

Reviewer #2: No

---

## [Editor Report · Acceptance letter]

27 Sep 2021

PONE-D-21-12334R1 

The epidemiology of atopic dermatitis in older adults: A population-based study in the United Kingdom 

Dear Dr. Abuabara:

I'm pleased to inform you that your manuscript has been deemed suitable for publication in PLOS ONE. Congratulations! Your manuscript is now with our production department. 

Kind regards, 

on behalf of

Dr. Dong Keon Yon 

Academic Editor

PLOS ONE